# Synthesis of Polylactic Acid Oligomers for Broad-Spectrum Antimicrobials

**DOI:** 10.3390/polym14204399

**Published:** 2022-10-18

**Authors:** Qi Bao, Ziheng Zhang, Baocheng Yu, Huize Sun, Polly Hang-mei Leung, Xiaoming Tao

**Affiliations:** 1Research Institute for Intelligent Wearable Systems, The Hong Kong Polytechnic University, Hong Kong 999077, China; 2School of Fashion and Textiles, The Hong Kong Polytechnic University, Hong Kong 999077, China; 3Wuhan Institute of Virology, Chinese Academy of Sciences, Wuhan 430071, China; 4University of Chinese Academy of Sciences, Beijing 100049, China; 5Department of Health Technology and Informatics, The Hong Kong Polytechnic University, Hong Kong 999077, China

**Keywords:** PLA, oligomer, IAV (influenza A virus), SARS-CoV-2 (COVID-19), antibacterial, antiviral, antifungal

## Abstract

Infectious microbial diseases are a major public health hazard, calling for more innovative antimicrobials. Herein, polylactic acid (PLA) oligomers have been explored and reported as a bio-safe and eco-friendly functional antimicrobial agent against pathogens, such as viruses (H1N1, H3N2, and SARS-CoV-2), bacteria (*E. coli*, *S. aureus*, *K. pneumoniae*, MRSA), and fungi (*C. albicans*). The PLA oligomers were prepared by direct catalyst-free condensation polymerization of l-lactic acid monomers and characterized by FT-IR and ^1^H-NMR. The antiviral results demonstrate that PLA oligomers possess robust (inhibiting rate > 99%) and rapid (<20 min) antiviral activity against two pandemic ssRNA viruses, including influenza A virus (IAV) and coronavirus (CoV). Furthermore, the PLA oligomers exhibit high antibacterial activities against both Gram negative (G^−^) and Gram positive (G^+^) bacteria. The PLA oligomers also perform efficiently in killing a large amount of *C. albicans* as high as 10^5^ cfu/mL down to zero at the concentration of 10 mg/mL. Thus, the broad-spectrum antimicrobial activity endowed the PLA oligomers with a promising biocidal option, except antibiotics in a wide range of applications, such as medical textiles, food preservation, water disinfection, and personal hygiene, in light of their unique biodegradability and biocompatibility.

## 1. Introduction

Infectious diseases caused by pathogens such as Middle East Respiratory Syndrome virus (MERS), Ebola virus, influenza A virus (IAV), severe acute respiratory syndrome (SARS) virus, Zika virus and the recent severe acute respiratory syndrome coronavirus disease virus (SARS-CoV-19) emerged as the most significant public health threat [1,2]. In addition, the ever-growing variation as well as the tolerance and resistance of pathogens toward common antibiotics and disinfectants has exacerbated the difficulty in treating infectious diseases and in the recovery of contaminated environments [3,4,5,6]. Thus, for the sake of securing the bio-safety of the whole society, there is an urgency to develop diverse antimicrobial agents with rapid and robust antimicrobial effects. Historically, certain metallic cations, such as silver and their nanoparticles/coatings, have been applied to fight against these infectious microbes. These Ag agents exhibit a multi-target mode of antimicrobial action, such as DNA damage, membrane breakage, reactive oxygen species (ROS) production, enzyme/protein deactivation, and interruption of metabolic pathways [7,8]. However, these Ag cations suffer from disadvantages, such as cytotoxicity and genotoxicity, as well as environmental hazards [9,10]. Thus, non-metallic compounds with good cytocompatiblity and environmental friendliness have also been required as alternatives for the implementation of antimicrobial agents.

Organic acids, for instance, acetic acid, succinic acid, cholic acid, and pyruvic acid, have been deployed as antimicrobial agents in abattoirs and hospitals and in the animal feed, food preservatives, personal washing products, and pharmaceutical industries to inhibit pathogenic micro-organisms in vitro for a long time [11]. Among these organic acid agents, lactate/lactic acid, the sugar metabolic product of lactic acid bacteria or yeasts, have drawn much more attraction. Lactic acid benefited from intrinsic advantages, such as green sustainability, high availability, good cell compatibility, and environmental friendliness. However, their poor processability as small molecules and their weak stability for long-term anchoring on a versatile surface posed another challenge, which limits their applications.

Meanwhile, some biological or synthetic polymers/macromolecules have displayed better opportunities, which have been recognized as novel antimicrobial agents [12]. Examples of these sorts of anti-infective polymers include antimicrobial peptide (AMP) [13], chitosan [14], *N*-halamines [15], Nylon-3 [16], etc. In spite of excellent antibacterial properties, these nitrogen containing polymers might accelerate the eutrophication of aqua systems and pose eco-hazards to aquatic life once leached into lakes and rivers after a long time. Furthermore, as we previously reported, a member of polyhydroxyalkanoates, poly (3-hydroxybutyrate) oligomers (PHB), was synthesized by ring-open polymerization of beta-butyrolactone, exhibiting a wide-spectrum antimicrobial activity. This kind of antimicrobial PHB polymer benefits from their benign environment without special concern of the secondary environmental risks induced by nitrogen and/or phosphorus elements because it contains only carbon, hydrogen, and three oxygen elements [17,18]. Furthermore, PHB oligomers have demonstrated other desirable features, such as biodegradability. However, the availability of the starting materials as well as the complexity of separation and purification currently limited these artificially synthesized PHB oligomers at the laboratory scale.

Thus, we proposed to oligomerize lactic acid, another kind of natural hydroxyl acid with higher availability, as an alternative to PHB oligomers, in the pursuit of antimicrobial agents in medical fabrics or plastics acting as the surface finishing agents or additives. Meanwhile, polylactic acid (PLA) has already been approved by the US Food and Drug Administration (FDA) for application in food and the human body mainly based on its biosafety. However, the antimicrobial function of PLA oligomers as a new kind of organic acid has not been well-known to our best knowledge, although that of lactic acid has been studied and reported for decades. Thus, the feasibility of PLA oligomers as an innovative and safe antimicrobial agent is being fully explored in this work. A facile one-step process, i.e., direct polycondensation of lactic acid in the absence of metal catalysts was applied to prepare the PLA oligomers. The broad-spectrum antimicrobial activity of the as-prepared PLA oligomers toward a wide variety of pathogenic micro-organisms including viruses, bacteria, and fungi were fully evaluated and reported in this work for the first time. The oligomerization of l-lactic acid makes PLA oligomers more advantageous compared to other competitors, such as AMP, chitosan, *N*-halamines, polybiguanides, etc. They are bio-safe, non-allergic, and eco-friendly with 100% biodegradability, with a residue that is merely CO_2_ and H_2_O. To this end, the ultimate impact of this work is the application of PLA oligomers as a cost-effective point-of-use microbicide for deactivating pathogens.

## 2. Materials and Methods

### 2.1. Synthesis of PLA Oligomers

Ultrapure l-lactic acid (≥85%) was obtained from J&K Chemical Co. Ltd. (Perlen, Switzerland) without any further purification. The l-lactic acid was added to the glass rotary evaporator (BUCHI rotavapor, Flawil, Switzerland), then rotated at a speed rate of 100 rpm, heated at an oil bath temperature of up to 180 °C, and maintained at the same conditions for over 4 h. The vacuum pressure in the evaporator was maintained in the range of 1 kPa~2 MPa during the reaction process. After the reaction was completed, the temperature of the reactor was allowed to return to room temperature naturally, the reaction products in the evaporating flask were then collected. The final yield of product was ~83.5%, calculated on weight.

### 2.2. Characterization of PLA Oligomers

The single-frequency decoupled proton nuclear magnetic spectroscopy (^1^H-NMR) spectrum was obtained on a ECZ500R 500 MHz Solid-State NMR spectrometer (JEOL, Tokyo, Japan) at room temperature. Five milligrams of each PLLA sample were dissolved in 0.6 mL deuterated chloroform (CDCl_3_, Sigma Aldrich, Saint Louis, MO, USA), which was then transferred into 5 mm NMR tubes. The NMR test was operated at 250.13 MHz (^1^H). The ^1^H chemical shifts are expressed in (*δ*). Eight scans were acquired by a 90° exciting pulse with an acquisition time of 5.1 s and a relaxation delay of 5 s. The spectral width was set to 6.4 kHz. The Fourier transform infrared (FT-IR) spectra were recorded on a Spectrum 100 spectrophotometer (Perkin Elmer, Waltham, MA, USA) with KBr pellets. The experimental conditions employed were as follows: scanning wavenumber from 4000 cm^−1^ to 650 cm^−1^; and 18 scans at 4 cm^−1^ resolution were obtained and averaged.

### 2.3. Assessment of Antibacterial/Antifungal Activity

The standard *Escherichia coli* (*E. coli*) ATCC 25922 and *Klebsiella pneumoniae* (*K. pneumoniae*) ATCC 4352 were used as the gram-negative model bacteria. The standard *Staphylococcus aureus* (*S. aureus*) ATCC 6538 and methicillin-resistant *Staphylococcus aureus* (MRSA) ATCC 43300 were used as the gram-positive model bacteria. The antibacterial activity of PLA oligomer was assessed according to ASTM E2149 13a-2015 standard (standard test method for determining the antimicrobial activity of immobilized antimicrobial agents under dynamic contact conditions). These bacteria suspensions were cultured overnight (12~18 h) with sterile Tryptic Soy Broth (Guangdong Huankai Microbial Sci. & Tech. Co., Ltd, Guangzhou, China) on a shaker at 37 °C with rotary agitation fixed at 150 rpm. Then, these culture media were diluted to 1.5~3.0 × 10^5^ CFU mL^−1^ with sterilized phosphate buffered saline (PBS) (BioUltra, J&K Scientific Ltd., Hong Kong) and then were incubated with the addition of PLA oligomer (10 mg mL^−1^) for 18 h at the culture conditions of 37 °C and 150 rpm agitation. The bacteria culture incubated without the PLA oligomer treatment served as their negative control group. Next, the bacteria cultures were diluted serially to 10^3^~10^1^ CFU mL^−1^ and 100 μL of sample were withdrawn from each diluted solution to spread over the solid agar plates. The solid agar plates were cultivated at 37 °C for 12~72 h. Then, the antibacterial rate was calculated by the following equation:(1)$$\mathrm{Antibacterial}\;\mathrm{rate},\;\%\left(\mathrm{CFU}/\mathrm{mL}\right)=\frac{N_{control}-N_{experimental}}{N_{control}}\times 100,$$
where *N_control_* and *N_experimental_* correspond to the average number of colony-forming units per milliliter calculated from the control without treatment and the experiment with the PLA treatment in triplicate, respectively.

Similarly, *Candida albicans* (*C. albicans*) ATCC No. 10231 was applied as the general fungus strain to investigate the antifungal ability of PLA oligomers based on the dynamic shake flask method for additional tests. The Sabouraud dextrose broth (Guangdong Huankai Microbial Sci. & Tech. Co.,Ltd, Guangzhou, China) was chosen as the culture media of the *C. albicans* tested.

The antimicrobial susceptibility (disk diffusion) tests in Muller–Hinton (M–H) agar plates were carried out according to the guidelines compiled by the Clinical and Laboratory Standards Institute (CLSI, Wayne, NE, USA). For the disk diffusion test, four kinds of model bacteria were spread onto the M-H agar plates with sterile cotton swabs from bacterial suspensions (10^6^–10^7^ CFU/mL). The PLA disk samples were placed at the center of the agar plate and pressed gently. After incubating at 37 °C for 48 h, the agar plates were observed for the presence of inhibitory zones surrounding the PLA disks. The diameter of the inhibitory zone was measured and then recorded.

The growth kinetics of bacteria treated with and without PLA oligomers was also detected and compared using a standard broth microdilution method. Briefly, 100 μL of the overnight culture broth (~10^6^ CFU mL^−1^) were transferred to the well of a 96-well polypropylene (Corning costar) plate, bacterial culture broth was mixed with 100 μL of PLA oligomers solution with a final concentration of 10 mg/mL. The plate was incubated at 37 °C. Bacterial growth was detected by OD_600nm_ at intervals of 2 h using a microplate reader (BMG LABTECH, Ortenberg, Germany).

### 2.4. Assessment of Virucidal Activity

The antiviral or virucidal effect of PLA oligomer was explored in vitro by the typical virus titer method, according to the technical standard for disinfection (2.1.1.10.7) of Ministry of Health, Beijing, P.R. China (2002) and ISO 21702:2019 standard (measurement of antiviral activity on plastics and other non-porous surfaces). In brief, the Madin-Darby canine kidney (MDCK) and Vero E6 cells have been selected as the host cells for the influenza A virus (IAV), such as H1N1, H3N2, and coronavirus (CoV), such as SARS-CoV-2, respectively. Both MDCK and Vero E6 cells were obtained from the American Type Culture Collection (ATCC) and cultured in Dulbecco’s modified Eagle’s medium (DMEM) (Gibco, Waltham, MA, USA) supplemented with 10% fetal bovine serum (FBS) and penicillin (100 U/mL) and streptomycin (100 μg/mL). Firstly, trypsinise (0.01% trypsinase in DMEM) the MDCK/Vero E6 host cells to prepare a single cell suspension of 2 × 10^5^ cells per ml in complete DMEM media. Dispense with the 8-channel pipette 100 μL of such suspension in each well of 96-well plates (2 × 10^4^ cellules/well). Then, dilute the virus suspension (H1N1, H3N2 and SARS-CoV-2) to be titrated in sterile tubes 10× from 10^−1^ to 10^−7^ (0.5 mL viral suspension in 4.5 mL of DMEM (without serum)). Dispense the different viral dilutions onto the cells with ten replicates for each dilution in 100 μL volumes. Such a process was repeated three times under the same conditions. Transfer these 96 well plates into an incubator at 37 °C supplied with 5% CO_2_ gas. Afterwards, the H1N1, H3N2, and SARS-CoV-2 virus were mixed and treated with equal volume PLA oligomer aqueous solution for different time intervals at 37 °C to obtain positive experimental group after which the PLA oligomer was removed by filtration/centrifugation. The negative control group was obtained by mixing the virus with equal volume ultrapure water and incubated for the same time intervals. The cell number of each plate well showing cytopathic effects (CPE) is observed and recorded to calculate the virus titer values after different time points post-infection. Afterwards, the titer values of control and experimental viruses were determined according to the standard TCID_50_ (TCID_50_, i.e., 50% tissue culture infectious dose which exhibits cytopathic effects (CPE) in 50% of a large number of inoculated cell culture wells) method [19,20].

Results were presented in both percent reductions (%) when measuring 50% tissue culture infective dose per millimeter (TCID_50_/mL) value and Log_10_ TCID_50_/mL reduction (KL) when calculating mean log_10_ density of TCID_50_ per millimeter.

Then, the virucidal rate was calculated by the following equation:(2)$$\mathrm{Antiviral}\;\mathrm{rate},\;\%\left({\mathrm{TCID}}_{50}/\mathrm{mL}\right)=\frac{N_{control}-N_{experimental}}{N_{control}}\times 100,$$
(3)$$\begin{array}{c} {\mathrm{Log}}_{10}\mathrm{virus}\;\mathrm{infectivity}\;\mathrm{titer}\;\mathrm{reduction}\;\left(\mathrm{KL}\right)\\ \;\;\;\;\;\;\;\;\;\;=\;{\mathrm{Log}}_{10}N_{control}-{\mathrm{Log}}_{10}N_{experimental} \end{array}$$
where *N_control_* and *N_experimental_* correspond to the average value of virus infectivity titer (TCID_50_/mL) calculated from the control without treatment and the experiment with the PLA treatment in triplicate, respectively.

## 3. Results and Discussion

Generally, the major methodology for the industrial production of PLA was the ring-opening polymerization (ROP) of lactide monomers catalyzed/initialized by certain transition metal complexes. However, the downstream isolation and separation of the residual metal catalysts, such as Sn(Oct)_2_, which is amenable to accumulate within the organisms once leaching from the final products, was highly challenging. Thereby, the facile direct linear self-condensation from ultrapure l-lactic acid precursors without the addition of metallic catalysts except trace water was chosen as the synthesis method of PLA oligomers, as shown in Figure 1.

The morphological and conformational properties of the as-prepared PLA oligomers were studied by ^1^H NMR and Fourier-transform infrared spectroscopy (FT-IR). As shown in the FT-IR spectra of PLA and l-lactic acid (Figure 2), a comparison of the FT-IR spectra of l-lactic acid before and after polymerization reveals three distinctive spectral differences in intensity of bands: the stretching vibrations of hydroxyl groups *v*(OH), the asymmetric stretching vibration of carboxyl groups *v*_as_(COO) and the deformation vibrations of hydroxyl groups *δ*(O−H). The typical stretching vibration peaks of O−H of hydroxyl groups were evident at 3406.2 cm^−1^ and 3493.0 cm^−1^, respectively. In a dried l-lactic acid monomer, the *v*(OH) band was centered at approximately 3406.2 cm^−1^ and its shape was sharpened. However, in the dried PLA oligomers, the *v*(OH) centered at approximately 3493.0 cm^−1^ shows a broad band as a result of hydrogen bond formation by the OH (sub)groups of carboxyl groups and alpha hydroxyl groups. The stretching vibrations of C=O were observed at 1760.8 cm^−1^ and 1738.8 cm^−1^, respectively.

The carbonyl index (CI) value was introduced to characterize the direct poly-esterification of l-lactic acid monomers. The CI value was calculated as the integrated peak area in the range from 1850 cm^−1^ to 1650 cm^−1^ divided by the integrated band absorbance of the methylene (CH_2_) scissoring peak from 1500 to 1420 cm^−1^ as shown in the following equation [21]:(4)$$\mathrm{Carbonly}\;\mathrm{index}\;\left(\mathrm{CI},\;\%\right)=\frac{Peak\;Area\;1890\sim 1520\;{\mathrm{cm}}^{-1}}{Peak\;Area\;1500\sim 1420\;{\mathrm{cm}}^{-1}\;}\times 100,$$

The according CI value of PLA oligomers was increased to 4.7 from 4.2 (LA) after the polymerization process due to the formation of the ester bonds. The ^1^H NMR spectrum of PLA oligomers is presented in Figure 3A, nearly all the protons exhibited well-defined resonances. The ^1^H NMR spectrum of PLA shows two sets of methyl H located in the range from 1.39 to 1.61 ppm, and the methine H of PLA appeared at 4.30–5.24 ppm. The peak separation of such protons confirmed the formation of PLA since the peak location of protons of repeating l-lactic units within main chains differs from that of chain ends. The most intense signals at 1.55 ppm and 1.44 ppm were assigned to CH_3_ protons and the downfield resonances (Peak D) were assigned to the repeating l-lactic units of backbone chain. The signals with a lower intensity at 5.18 ppm and 4.36 ppm were attributed to C−H protons and the downfield resonances (Peak B) were attributed to the chain end-units adjacent to the terminal carbonyl groups. The Peak B was typically recognized as a characteristic quintet, which was split by adjacent one terminal O−H proton of carboxylic acid groups and three C−H protons of methyl groups. The split peaks located at 4.31, 4.33, 4.34, 4.35, 4.37, and 4.38 ppm, respectively. This also demonstrated the molecular charity of the PLA oligomers has been well maintained after polymerization since racemization of PLA generally occurs at temperatures exceeding 200 °C. The molar mass (*M*_n_) of the PLA oligomer was also estimated by dividing the integrated area of the methine protons of the repeating unit by that of the methine protons of the end-units. In this way, the *M*_n_ of poly(l-lactic acid) was calculated to 450.5 (*n* ≈ 6), following into the medium chain range. As seen from Figure 3B, the as-synthesized PLA oligomers were in the state of viscous liquid being almost transparent light yellow.

As two subtypes of the *Orthomyxoviridae* family, human (H3N2 and H1N1) influenza A viruses (IAVs) cause severe respiratory diseases [22,23]. As listed in Table 1 and Table 2, the PLA oligomers exhibited negligible cytotoxicity against MDCK host cells at concentrations up to 20 mg/mL, indicating a safe profile in the following study. The PLA oligomers clearly displayed a rapid but excellent inhibition effect against influenza A virus infection, confronting both A/Puerto Rico/8/34 ATCC VR-1469 (H1N1) virus and A/Aichi/2/68 ATCC VR-1679 (H3N2) virus. The virucidal rate of PLA oligomers against H1N1 and H3N2 virus within 2 h could reach >99.99% after IAV infection. In particular, such a high virucidal rate could be achieved within 10 min in the case of the H3N2 virus. The TCID_50_ value of treated IAV virus was decreased by at least four orders of magnitude, as compared with the negative control group. The influenza virus takes 8~10 h to accomplish one life cycle. The virus-killing of PLA oligomers can be accomplished within one life cycle of influenza virus. The virucidal rate against H1N1 and H3N2 virus was >99%, and the TCID_50_ value was reduced by two orders of magnitude within 10 min after the detoxification treatment. This also indicated that the virus-killing behavior of PLA oligomers was time-dependent.

As one of the severe acute respiratory syndrome coronavirus (SARS-CoV) species in the *Coronaviridae* family, the SARS-CoV-2 virus and its variants were the chief culprits responsible for the current outbreak of the coronavirus disease (COVID-19) pandemic, which resulted in over 6 million deaths worldwide so far [24,25]. Thereafter, the antiviral effectiveness of PLA oligomers was tested against the SARS-CoV-2 virus and the results were listed in Table 3 and Table 4. Firstly, negative control Vero E6 host cells without the inoculation of virus seeds grow well after 1.5 h of incubation, indicating PLA had no/less toxic effect on the host cells at the concentration of 20 mg/mL PLA solution. Normally, the SARS-CoV-2 virus could survive as long as 7.96–10.2 h on the surfaces of stainless steel/glass/plastic, much longer than that of IVA (1.65–2.00 h). The titer value of SARS-CoV-2 virus was reduced by four orders of magnitude from 9.68 × 10^6^ (TCID_50_/mL) to lower than 3.16 × 10^2^ (TCID_50_/mL) within 1.5 h at the presence of 20 mg/mL PLA oligomers, indicating an excellent antiviral capability of PLA oligomers. The antiviral rate against SARS-CoV-2 virus of PLA oligomers could be >99.99% within 1.5 h. Meanwhile, PLA oligomers also displayed rapid virus inhibiting capability to reduce the transmitting risk of SARS-CoV-2 virus. The virus titer (TCID_50_/mL) value was reduced by 99.99% when the contact time was cut to 20 min. Although the dose of PLA oligomers has been diluted by half to ~10 mg/mL, no significant reduction was observed and the antiviral rate of 99.99% could also be reached within 20 min. It has been shown that the as-prepared PLA oligomers effectively inhibit both IVA and SARS-CoV-2.

In addition, the antibacterial susceptibility tests were carried out to confirm the antibacterial capability of PLA oligomers based on a micro-disk diffusion method. The antibacterial activity of PLA oligomers was compared at the presence of four representative bacteria of Gram-negative bacteria and Gram-positive bacteria, that is, *E. coli*, *K. pneumonia*, *S. aureus*, and MRSA. The clear growth-inhibition zone around each of the paper disks loaded with PLA oligomers was well-identified after incubation for *E. coli*, *K. pneumonia*, *S. aureus*, and MRSA (Figure 4). The inhibiting diameter was in the range of 1.65–2.05 cm with the loading densities between 0.60 to 0.65 mg/mm^2^ as summarized in Table 5. The results of the disk diffusion test have evidenced qualitatively that both the Gram-negative and Gram-positive strains are susceptible to the PLA oligomers.

To further evaluate the antibacterial effect, PLA oligomers were added into bacteria (*S. aureus*) containing M-H broth. The level of survival was examined by the plate counting method after shaking incubation at 37 °C for 2 h and 12 h, respectively. Representative agar plates were prepared from undiluted solutions of the negative control experiment and positive experimental at the presence of PLA oligomers (Figure 5). The control group showed a robust growth of *S. aureus* cells after 12 h. By contrast, for the PLA oligomer samples, no distinguishable bacterial growth was observed, and the bacteria cells of *S. aureus* in the presence of PLA oligomers reduced greatly.

To further examine the antibacterial effect of PLA oligomers, typical growth kinetics analysis of *S. aureus* treated with and without PLA oligomers was also performed. As shown in Figure 6, the negative control groups of *S. aureus* experienced a rapid logarithmic growth before reaching the stationary stage. By contrast, the bacteria cell growth was depressed at the presence of PLA oligomers because no increase or rebound of OD_600_ value was found. This result indicated that PLA oligomers could completely inhibit the growth of *S. aureus* within 12 h at a concentration of 10 mg mL^−1^.

The as-synthesized PLA oligomers were tested for their antibacterial activity against four bacteria strains including *E. coli*, *S. aureus*, *K. pneumoniae*, and MRSA. The detailed results of antibacterial rates are listed in Table 6. The PLA oligomers have been evidenced to process a broad-spectrum inhibitory effect against both Gram-negative strains, such as *E. coli* and *K. pneumoniae*, and Gram-positive strains, such as *S. aureus* and MRSA. All of the three antibacterial reduction rate of PLA oligomers achieved was >99.99%, even at a lower concentration of 10 mg/mL. As the variant (sub) species of *S. aureus*, MRSA is notorious for their superior resistance to all current antibiotics. Nevertheless, PLA oligomers could inactivate a majority of MRSA with the antibacterial reduction rates of 80.73% and 84.44% at a concentration of 10 mg/mL and 20 mg/mL, respectively. This also provided a treatment option for MRSA infections.

*C. albicans* is one of the major causes of systemic fungal infections in the immunocompromised patients. Thus, *C. albicans* was deployed to evaluate the antifungal susceptibility of PLA oligomers. PLA oligomers were added into *C. albicans* containing broth culture. Similarly, the level of survival of *C. albicans* was examined by the plate counting method after the shaking incubation process. Agar plates were captured from undiluted solutions of the control group and experimental samples. As we can see from Figure 7, the control experiments in the absence of PLA oligomers showed a robust growth of *C. albicans*. For the experimental samples, strong antifungal activity was observed, and no fungal cell in the presence of the PLA oligomers survived. The detailed antifungal rate of PLA oligomers against *C. albicans* was calculated to 84% and 96% with respect to the dose of 10 mg/mL and 20 mg/mL, respectively. This is due to the two-layered cell-well structure of *C. albicans* providing better protection from the attacks of chemical disinfectants [26]. Additionally, the fungal-killing ability of PLA oligomers was demonstrated to be dose-dependent.

Although organic acids have long been applied as food additives and preservatives, the underlying antimicrobial modes of action are still not fully understood. Their antimicrobial activity may also vary as the change of the physiological status of micro-organisms and the physicochemical characteristics of the local in vitro circumstances. Theses organic acids can act either as a source of carbon and energy metabolized by micro-organisms, or as inhibitory agents to suppress the growth of micro-organisms, depending on organic acids, such as concentration, chain length, and micro-organisms. Herein, some possible antimicrobial mechanisms on PLA oligomers have been tentatively explained and discussed as shown in Figure 8. (i) Firstly, the cell membrane was identified as the primary target of action. Wang et al. discovered the disruptive effect of lactic acid on the integrity of the cytoplasmic membrane as one of the major antibacterial modes of LA based on TEM observation [27]. They also pointed out some membrane proteins were acid-sensitive and that membrane permeability interfered with organic acids. Leakage of cellular proteins occurs and this has also been confirmed by gel electrophoresis in the case of PHB oligomers. Indeed, some ion channels were acid-sensitive, such as TRPV1, which were gated by severe acidosis. In addition, recent studies revealed the entire outer surface of *E. coli* bacteria was spread by a porous network connected by abundant beta-barrel outer-membrane proteins, such as the trimeric OmpF porin [28]. Thus, the membrane protein played a by far more role vital in the stabilization of the cell membrane than we thought. The denaturation of membrane protein or structural protein resulted in the severe damage or lysis of the cell membrane structure. Such a membrane protein damage statement could reasonably explain the antiviral activity of PLA oligomers against two types of enveloped viruses, including the SARS-CoV-2 and influenza viruses because these envelope proteins are essential for the entry of viruses into host cells. (ii) Secondly, the activity of respiratory enzymes could be decreased by intracellular acid pH values, and this may be a secondary effect of the acidification of the cytoplasm. For example, the ATP levels in *B. subtilis* and *E. coli* were reduced after organic acids exposure, indicating the energy metabolism was interrupted by the uncoupling electron transport capabilities of undissociated acid [29]. Organic acids could react with the thiol group of cysteine residues of these related enzymes, and the metabolic disorders of the cells were further accelerated [30]. (iii) Finally, the PLA oligomer could cause physical damage to the DNA macromolecule. Meanwhile, the DNA syntheses and replication process were both inhibited since the DNA polymerase activity was depressed. Another possibility is that the acid anion may interfere with the conformation of the DNA molecule by interacting with ion charges around it. For example, the DNA synthesis in *E. coli* was seen to be depressed to the largest degree at the presence of both formic and propionic acids by in vitro experiments [31]. Thereafter, the multiple possibility of the damage of proteins and DNA associated with the cells ultimately withering exists as the presence of PLA oligomers might elicit such exceptional broad-spectrum antimicrobial performance.

Consequently, the intrinsic characteristics and flexible processability of oligomers enabled PLA oligomers to be easily deposited on versatile surface through functional coating/finishing process as shown in Figure 9. The hydrophilicity, low cost, biocompatible, eco-friendliness and cost-effectiveness of PLA oligomers makes it reasonable to anticipate the large-scale applications in biomedical devices (e.g., wound dressing), hygiene cleaning (e.g., hand/body wash cream), personal protective equipment (e.g., facemasks) and beverage packaging (e.g., seafood/fruit preservative bag). This kind of PLA oligomers material can also be applied but not limited in practical fields such as disinfectant, antimicrobial finishing agents, hospital supplies, hygiene products, public health, etc. For example, these as-synthesized PLA oligomers have been coated over medical facemasks by us and evaluated by over 2000 end-users including the elderly people (age > 60). Most of the feedback was satisfied and positive in actual practice and no adverse effects have been found to date.

## 4. Conclusions

In the present study, a broad-spectrum antimicrobial agent, PLA oligomer, was synthesized and discovered through a series of antimicrobial tests against viruses, bacteria, and fungi for the first time. We found that PLA oligomers exhibited strong antiviral effects against coronavirus (SARS-CoV-2) as well as human influenza viruses, including H1N1 and H3N2 two subtypes, and exhibited low toxicity against both MDCK and Vero cells in vitro. The antibacterial activity of the prepared PLA oligomers was confirmed by the disk diffusion method against Gram-negative *E. coli* and *K. pneumoniae* strains and Gram-positive *S. aureus* and MRSA strains. The antibacterial activity of PLA oligomers has also been demonstrated against *E. coli*, *S. aureus*, *K. pneumoniae*, and MRSA with antibacterial rate values of >99.99%, >99.99%, >99.99%, and >80.73% at a concentration of 10 mg/mL. The fungicide action against *C. albicans* was determined with an antifungal activity of 96% at the concentration of 20 mg/mL. Future studies would focus on the action mode of PLA oligomers at in vitro and in vivo levels.

## Figures and Tables

**Figure 1 polymers-14-04399-f001:**
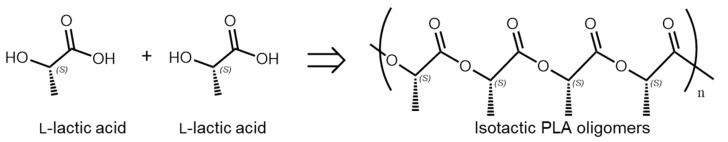
Schematic illustration of the PLA oligomers polymerized by direct poly-condensation of l-lactic acid monomers.

**Figure 2 polymers-14-04399-f002:**
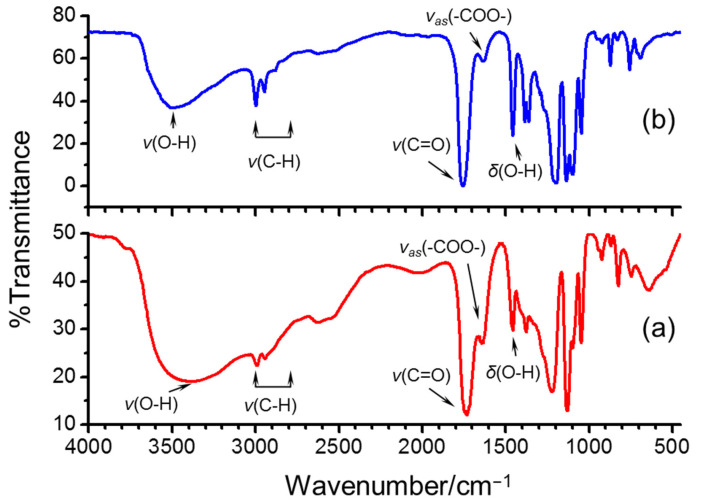
FT-IR transmission spectra of (**a**) l-lactic acid and (**b**) as-prepared PLA oligomers.

**Figure 3 polymers-14-04399-f003:**
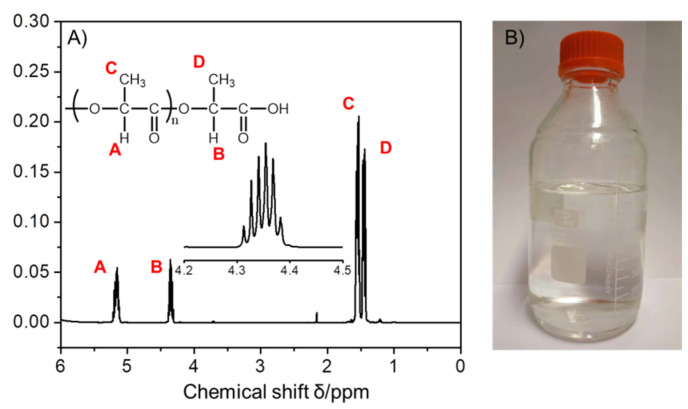
(**A**) ^1^H NMR spectrum of the as prepared PLA oligomers in CDCl_3_. Inset:  corresponding PLA chemical structure and enlarged partial spectrum in a range from 4.2 to 4.5 ppm and (**B**) digital image of the as prepared PLA oligomers.

**Figure 4 polymers-14-04399-f004:**
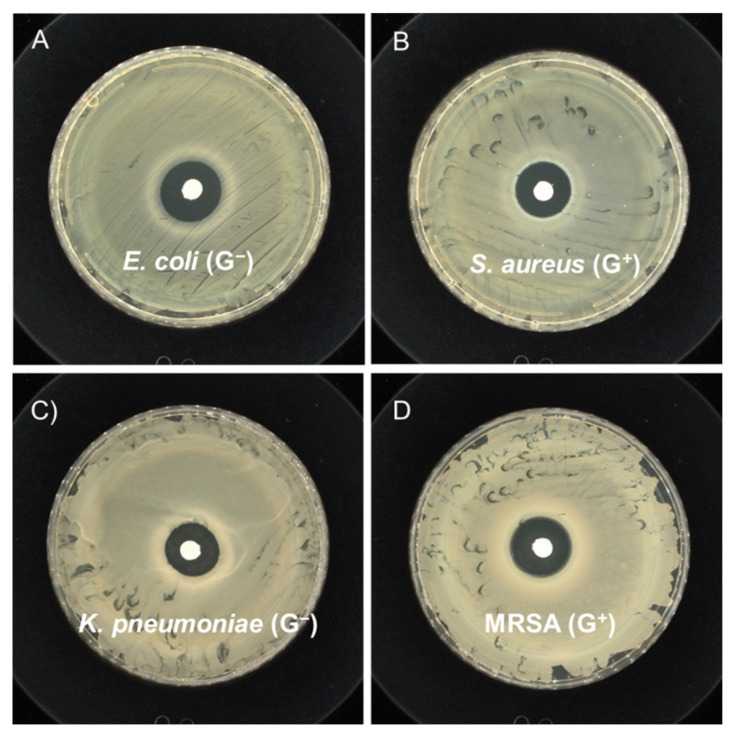
Kirby–Bauer antibiotic strength measurements for pure solid state PLA disks against (**A**) *E. coli* (ATCC No. 25922), (**B**) *S. aureus* (ATCC No. 6538) (**C**) *K. pneumoniae* (ATCC No. 4352) and (**D**) MRSA (ATCC No. 43300).

**Figure 5 polymers-14-04399-f005:**
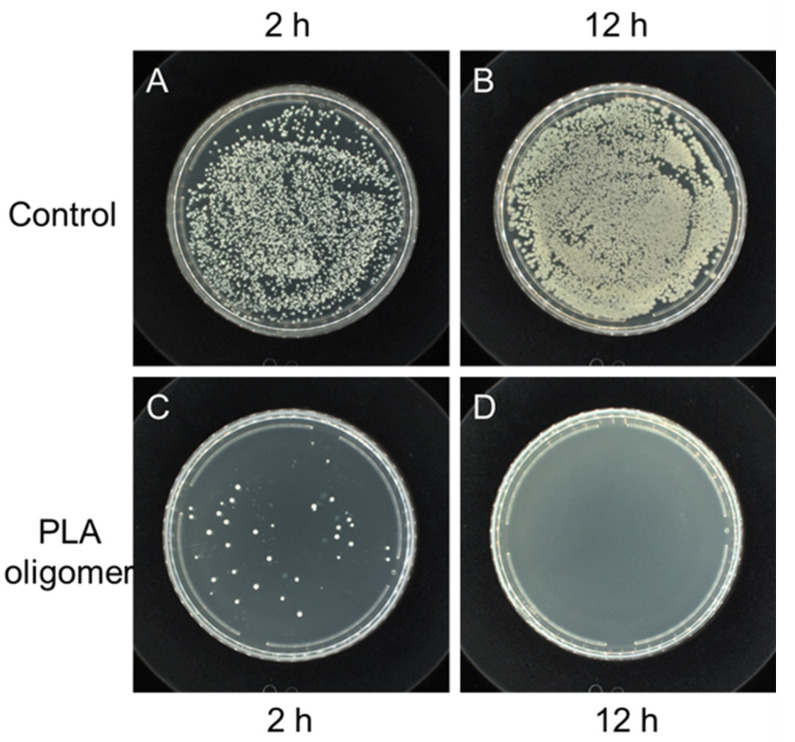
Antibacterial tests against *S. aureus* (ATCC No. 6538). Digital photographs of colonies of *S. aureus* cultivated on agar plates obtained from bacteria cell culture suspensions with PBS buffer solutions (Control) after (**A**) 2 h’ and (**B**) 12 h’ incubation at 37 °C and PLA oligomers after (**C**) 2 h’ and (**D**) 12 h’ incubation at 37 °C.

**Figure 6 polymers-14-04399-f006:**
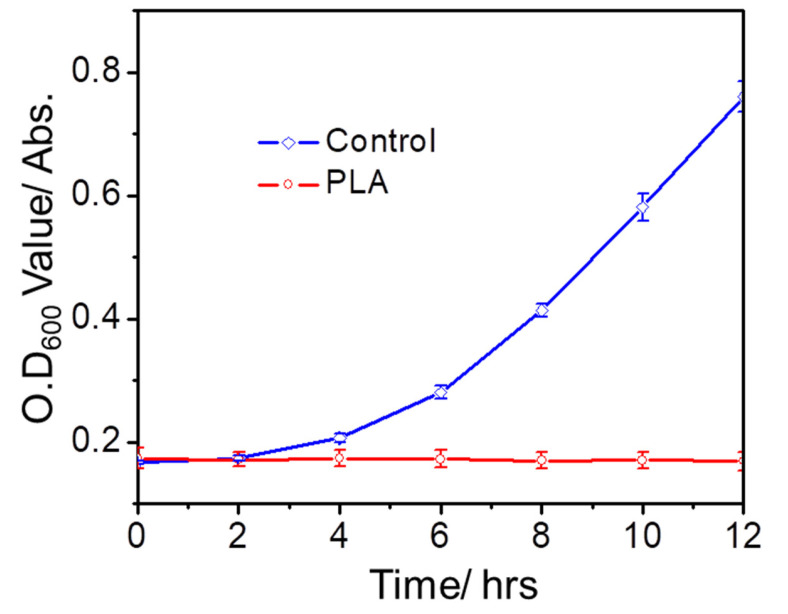
The OD600 values of *S. aureus* after treatment with and without (Control) PLA oligomers (10 mg/mL) at different time intervals.

**Figure 7 polymers-14-04399-f007:**
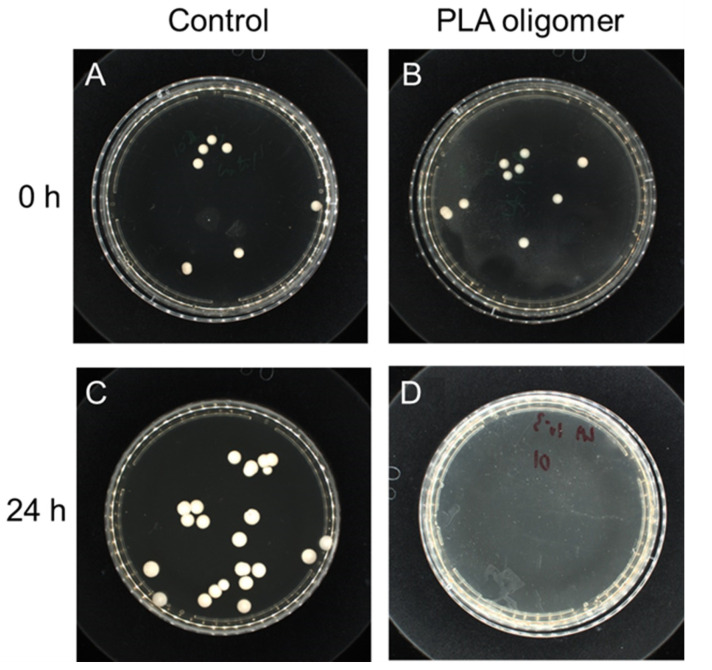
Antifungal tests against *C. albicans* (ATCC No.10231). Digital photographs of colonies of *C. albicans* cultivated on agar plates obtained from bacteria cell culture suspensions with PBS buffer solutions (Control) after (**A**) 0 h’ and (**B**) 24 h’ incubation at 37 °C and PLA oligomers after (**C**) 0 h’ and (**D**) 24 h’ incubation at 37 °C.

**Figure 8 polymers-14-04399-f008:**
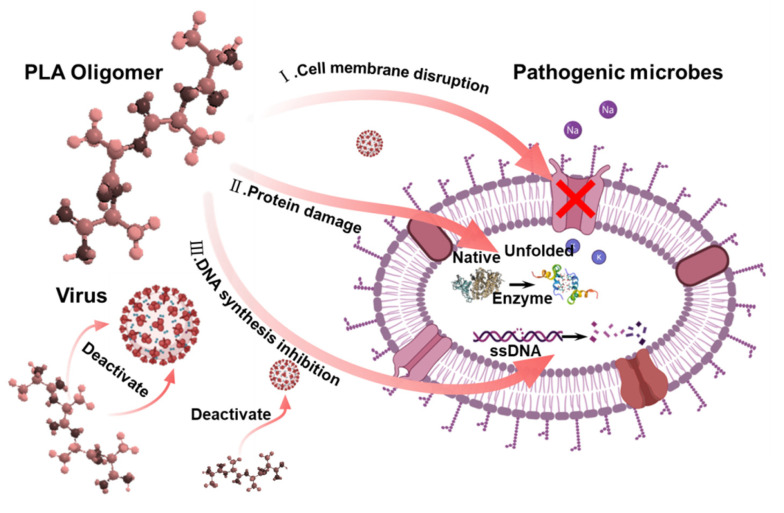
Schematic illustration of the in vitro and in vivo antimicrobial mechanism of the PHB oligomers.

**Figure 9 polymers-14-04399-f009:**
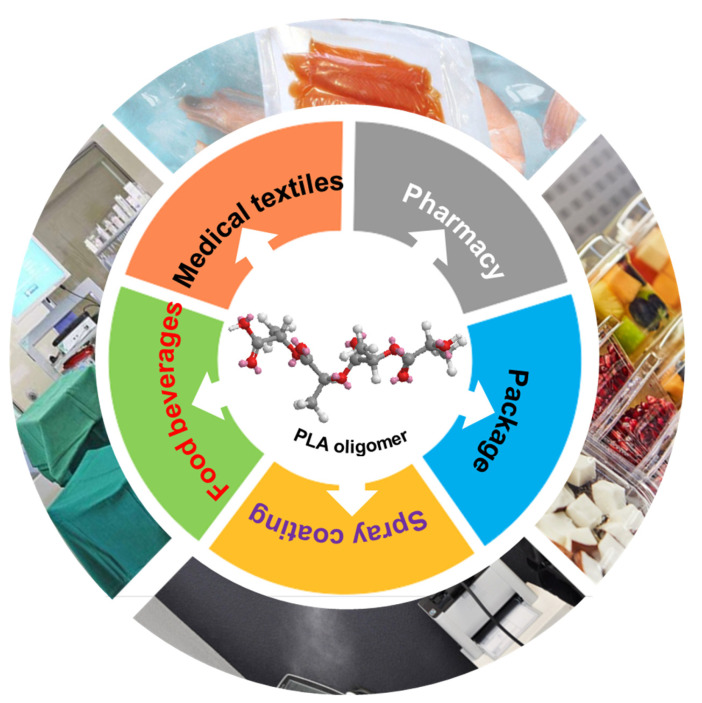
Potential applications of PLA-based antimicrobial oligomers.

**Table 1 polymers-14-04399-t001:** Antiviral performance of PLA oligomers against *Influenza A* H1N1 virus.

Virus Species and Host Cell	Culture Concentration	Contact Time	Groups	LgTCID_50_/mL	Average TCID_50_/mL	Average LgTCID_50_/mL	Logarithm Reduction Value (KL)	Virus Inactivation Ratio (%)
Virus: *Influenza A virus*	0 mg/mL	1 h	Ctrl 1	5.80	5.89 × 10^5^	5.77	-	-
			Ctrl 2	5.80				
			Ctrl 3	5.71				
H1N1 (ATCC VR-1469)	20 mg/mL	1 h	PLA1	<1.50	<31.6	<1.50	>4.27	>99.99
			PLA2	<1.50				
			PLA3	<1.50				
Host cell: MDCK	20 mg/mL	1 h	PLA1D	1.57	39.81	1.60	4.17	>99.99
			PLA2D	1.67				
			PLA3D	1.57				

Negative control MDCK host cells without the inoculation of virus seeds grow well.

**Table 2 polymers-14-04399-t002:** Antiviral performance of PLA oligomers against *Influenza A* H3N2 virus.

Virus Species and Host Cell	Culture Concentration	Contact Time	Groups	LgTCID_50_/mL	Average TCID_50_/mL	Average LgTCID_50_/mL	Logarithm Reduction Value (KL)	Virus Inactivation Ratio (%)
Virus: *Influenza A virus*	0 mg/mL	10 min	Ctrl 1	5.90	6.85 × 10^5^	5.83	-	-
			Ctrl 2	5.80				
			Ctrl 3	5.80				
H3N2 (ATCC VR-1679)	20 mg/mL	10 min	PLA1	<1.50	<31.6	<1.50	>4.33	>99.99
			PLA2	<1.50				
			PLA3	<1.50				
Host cell: MDCK	20 mg/mL	10 min	PLA1D	3.00	1.17 × 10^3^	3.07	2.76	99.83
			PLA2D	3.20				
			PLA3D	3.00				

Negative control MDCK host cells without the inoculation of virus seeds grow well.

**Table 3 polymers-14-04399-t003:** Antiviral performance of PLA oligomers against SARS-CoV-2 virus.

Virus Species and Host Cell	Culture Concentration	Contact Time	Groups	TCID_50_/mL	Average TCID_50_/mL	Average LgTCID_50_/mL	Logarithm Reduction Value (KL)	Virus Inactivation Ratio (%)
Virus: SARS-CoV-2 Host cell: Vero E6	0 mg/mL	1.5 h	Ctrl 1	1.78 × 10^7^	9.68 × 10^6^	6.99	-	-
			Ctrl 2	5.62 × 10^6^				
			Ctrl 3	5.62 × 10^6^				
	10 mg/mL	1.5 h	PLA1	<3.16 × 10^2^	<3.16 × 10^2^	<2.50	>4.49	>99.99
			PLA2	<3.16 × 10^2^				
			PLA3	<3.16 × 10^2^				
	20 mg/mL	1.5 h	PLA1	<3.16 × 10^2^	<3.16 × 10^2^	<2.54	>4.49	>99.99
			PLA2	<3.16 × 10^2^				
			PLA3	<3.16 × 10^2^				

Negative control Vero E6 host cells without the inoculation of virus seeds grow well.

**Table 4 polymers-14-04399-t004:** Antiviral performance of PLA oligomers against SARS-CoV-2 virus.

Virus Species and Host Cell	Culture Concentration	Contact Time	Groups	TCID_50_/mL	Average TCID_50_/mL	Average LgTCID_50_/mL	Logarithm Reduction Value (KL)	Virus Inactivation Ratio (%)
Virus: SARS-CoV-2 Host cell: Vero E6	0 mg/mL	20 min	Ctrl 1	3.16 × 10^7^	1.89 × 10^7^	7.28	-	-
			Ctrl 2	2.04 × 10^7^				
			Ctrl 3	4.64 × 10^6^				
	10 mg/mL	20 min	PLA1	<3.16 × 10^2^	<3.16 × 10^2^	<2.50	>4.78	>99.99
			PLA2	3.98 × 10^2^				
			PLA3	<3.16 × 10^2^				
	20 mg/mL	20 min	PLA1	<3.16 × 10^2^	<3.16 × 10^2^	<2.54	>4.78	>99.99
			PLA2	<3.16 × 10^2^				
			PLA3	<3.16 × 10^2^				

Negative control Vero E6 host cells without the inoculation of virus seeds grow well.

**Table 5 polymers-14-04399-t005:** Kirby–Bauer antibacterial strength results of PLA oligomers against various bacteria.

PLA Oligomer Parameters	*E. coli* (G^−^)	*S. aureus* (G^+^)	*K. pneumoniae* (G^−^)	MRSA (G^+^)
Loading density (mg/mm^2^)	0.64	0.60	0.60	0.64
Inhibiting diameter (cm)	1.90 ± 0.15	1.85 ± 0.15	1.70 ± 0.05	1.80 ± 0.12

**Table 6 polymers-14-04399-t006:** Antibacterial rates of PLA oligomers against various bacteria.

PLA Oligomer Concentration	*E. coli* (G^−^)	*S. aureus* (G^+^)	*K. pneumoniae* (G^−^)	MRSA (G^+^)
10 mg/mL	>99.99%	>99.99%	>99.99%	>80.73%
20 mg/mL	>99.99%	>99.99%	>99.99%	>84.44%

## Data Availability

Not applicable.

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
