# Peer review of "Synthesis of Polylactic Acid Oligomers for Broad-Spectrum Antimicrobials"

_polymers, 2022, doi:10.3390/polym14204399_

Round 1
Reviewer 1 Report
The manuscript describes an anti-microbial synthetic PLA oligomers important anti-G+,G-and C. albicans, damaging the integrity of the cell membrane. The objective is clear. The results are consistent with the discussion. Accordingly, the conclusions can be understood. The statistics is very well done.However, the following questions should be considered before acceptance for publication.
1) In the introduction, the authors should emphasize the novelty of the work.
2) The synthetic PLA oligomers should be described in detail, its application range should also be provided.
3) The author should provide the microstructure (SEM) analysis data after antibacterial treatment.
Author Response
Response to Editor and Reviewers’ Comments
We highly appreciate the editor and reviewers for your valuable time and effort in reviewing and commenting our manuscript. We think these suggestions are very helpful for our present and future work. We have carefully considered the all comments and revised our manuscript accordingly. All the revisions are highlighted in the revised manuscript for easy reference. The detailed comments and responses are listed as below:
Reviewer #1:
Comments: The manuscript describes an anti-microbial synthetic PLA oligomers important anti-G+,G-and C. albicans, damaging the integrity of the cell membrane. The objective is clear. The results are consistent with the discussion. Accordingly, the conclusions can be understood. The statistics is very well done. However, the following questions should be considered before acceptance for publication.
1) In the introduction, the authors should emphasize the novelty of the work.
Response: Thanks for this reviewer’s comments. We have considered all reviewers’ comments and revised our manuscript carefully. In terms of the novelty of this work, on the one hand, we firstly discovered the excellent wide-spectrum antimicrobial properties of PLA oligomer, which could be effective to kill gram-positive bacteria, gram-negative bacteria, fungi, and coronaviruses. In particular, the antiviral activity of as-synthesized PLA oligomers has been founded and reported within this work for the first time. On the other hand, compared with other antimicrobial agents (e.g. nano-silver, polybiguanides, chitosan etc.), this type of oligomer material was benefited from their intrinsic advantages like being safe, non-allergic, low carbon emission and fully biodegradable with residue of only CO2 and H2O.
According to the reviewer’s suggestion, the last paragraph of the introduction section has been modified. The novelty of this work has been highlighted in the revised manuscript accordingly (See Page 2/17).
2) The synthetic PLA oligomers should be described in detail, its application range should also be provided.
Response: Thanks for the reviewer’s comments. According to the reviewer’s suggestion, the first paragraph of the Materials and Methods section has been revised for clarity (See Page 2/17 ~ 3/17). Additionally, the application of the PLA oligomers has been expanded and added in the revised manuscript according to the reviewer’s suggestion (See Page 14/17).
3) The author should provide the microstructure (SEM) analysis data after antibacterial treatment.
Response: Thanks for the reviewer’s concern. Indeed, it’s better for us to get insight into the antimicrobial mechanism/effects of the PLA oligomers on a microscopic scale. For clarity, the TEM observation was carried out since the resolution of TEM was better than that of SEM. However, that’s a long story on the studies of the molecular mechanism of the antimicrobial performance of PLA oligomers, which will be carried out systematically in the following days. That study is still on-going. However, we could also introduce the preliminary results of TEM observation briefly herein. There was no significant change on the shape/size of the bacteria (e.g., E.coli) after treatment with PLA oligomers within half an hour. However, the most distinctive difference might lie in the amount of bacteria and it seems that the growth of bacteria has been depressed at the presence of PLA oligomers.
Fig.1 TEM images of A), B) negative control (untreated) of E.coli bacteria
and C), D) PLA treated E.coli bacteria

Reviewer 2 Report
The manuscript "Synthesis of Polylactic Acid Oligomers for Broad-spectrum Antimicrobials" describe the application of polylactic acid (PLA) oligomers as antimicrobial agents against a series of pathogens.
Before it can be accepted for publication, the manuscript needs to address several issues:
1. In the Introduction chapter, the manuscript needs to focus on the application of PLA oligomers, as emerging from literature data.
2. The NMR and FTIR analyses are useful for the characterization of the oligomers. However, I would suggest the use of other analytical methods, such as thermal analyses or electron microscopy.
3. A positive control should be used for all the antimicrobial tests
Author Response
Response to Editor and Reviewers’ Comments
We highly appreciate the editor and reviewers for your valuable time and effort in reviewing and commenting our manuscript. We think these suggestions are very helpful for our present and future work. We have carefully considered the all comments and revised our manuscript accordingly. All the revisions are highlighted in the revised manuscript for easy reference. The detailed comments and responses are listed as below:
Reviewer #2:
Comments: The manuscript "Synthesis of Polylactic Acid Oligomers for Broad-spectrum Antimicrobials" describe the application of polylactic acid (PLA) oligomers as antimicrobial agents against a series of pathogens.
Before it can be accepted for publication, the manuscript needs to address several issues:
- In the Introduction chapter, the manuscript needs to focus on the application of PLA oligomers, as emerging from literature data.
Response: Thanks for the reviewer’s comments. We have appreciated it so much. In the revised introduction section, we emphasized the novelty of the PLA oligomers (Page 2/17). Meanwhile, we emphasized and expanded the application of PLA oligomers in the last paragraph of the results and discussion section, as seen on Page 14/17.
- The NMR and FTIR analyses are useful for the characterization of the oligomers. However, I would suggest the use of other analytical methods, such as thermal analyses or electron microscopy.
Response: Thanks for the reviewer’s comments. Both the thermal analyses and electron microscopy are good analytical methods to investigate the PLA polymers. However, the fact is that the as-synthesized PLA oligomers in this work was in the state of viscous liquid, which contains trace water, which was not applicable for the thermal analysis like TGA and DSC as well as SEM and TEM analysis. In order to make it clear, we added the digital image of the liquid sample prepared in Fig.3 in the revised manuscript (See Page 6/17).
- A positive control should be used for all the antimicrobial tests
Response: Thanks for the reviewer’s professional suggestion. In most cases of this work, both the control and the experimental groups have been compared together under the parallel conditions. However, in the qualitative analysis of the antibacterial activity of PLA oligomers, i.e., the disk diffusion experiment, the positive control group was not included. This is because the loading concentration on the paper disk was difficult to control as a result of the viscous liquid state of PLA oligomers. The quantitative analysis was not applicable in this case based on the disk diffusion experiments. However, the clear inhibiting zone appeared indicate qualitatively that the antibacterial ability of PLA oligomers did exist in all the four cases even without positive control groups. Meanwhile, we further carried out the antibacterial performance quantitatively according to the ASTM standard as well as the CLSI guideline to evaluate the antibacterial ability of PLA oligomers. The control groups have been included in the following part.
